# Research on the Sustainable Development and Dynamic Capabilities of China's Aircraft Leasing Industry Based on System Dynamics Theory

**Weiwei Lin \*, Jing Lu** 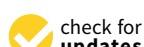**, Jinfu Zhu and Li Xu**

College of Civil Aviation, Nanjing University of Aeronautics and Astronautics, 29 Jiangjun Avenue, Jiangning District, Nanjing 211106, China; lj_ppx@nuaa.edu.cn (J.L.); zhujf@nuaa.edu.cn (J.Z.); xuli01@szairport.com (L.X.)
\* Correspondence: linww@cdb-leasing.com

**Abstract:** The Boeing 737 MAX crisis and COVID-19 pandemic have seriously influenced the development of China's aircraft leasing industry in the past two years. This paper applies system dynamics theory to explore the sustainable development of China's aircraft leasing industry. It analyses the dynamic mechanism and constructs a system dynamics model. Based on China's macroeconomic data and historical data from the financial, aviation, and leasing industries, it aims to stimulate the development of China's aircraft leasing industry in the next five years. Through sensitivity analysis, this research finds that changes in GDP growth have the most obvious impact on the sustainable development of China's aircraft leasing industry. Reducing the average financing cost and the income tax rate of aircraft leasing companies, increasing their investment in talent, and controlling risk will increase the market share of China's aircraft leasing companies and promote the development of the industry. However, increasing the number of aircraft leasing companies has little effect on market share. On this basis, this paper proposes policy recommendations to promote the sustainable development of China's aircraft leasing industry.

**Keywords:** complex system; aircraft leasing industry; system dynamics; sustainable development; simulation

## 1. Introduction

Leasing is one of the most vibrant and dynamic industries in the world. From an economic perspective, leasing can be defined as "a contract between two parties where one party (the lessor) provides an asset for usage to another party (the lessee) for a specified period of time, in return for specified payments" [1]. It facilitates the financing of equipment and real property. It fosters economic growth, creates employment, and enhances tax revenue. Leasing affects many spheres of human lives as it encompasses cars, furniture, airplanes, restaurant equipment, computers, telecom equipment, medical equipment, and many other properties [2]. It plays an important role in the promotion of a country's economic development [3]. The aircraft leasing industry is closely related to the aviation manufacturing, aviation transportation, and financial industries. It involves civil aircraft manufacturers, aircraft leasing demand subjects, supply subjects, regulatory subjects, trading markets, institutional environments, and other elements. The sustainable development of the aircraft leasing industry is an organically unified change process of the quantitative growth and qualitative development of the core elements of aircraft leasing [4]. In this process, the capabilities of civil aircraft manufacturing companies continue to improve. Aircraft leasing companies enrich leasing products, improve professional management capabilities, and meet the development needs of the aviation industry. At the same time, the aircraft leasing business model continues to improve, the aircraft leasing market environment continues to be optimized, and the scale of aircraft leasing transactions continues to expand. Promoting the sustainable development of the aircraft leasing industry is of great

practical significance for improving China's aviation service capabilities and emergency dealing capability, as well as optimizing resource allocation. To accelerate the development of the aircraft leasing industry, China has proposed a "three-step" plan. From 2020 to 2030, it will build an aircraft leasing industry cluster and form several internationally competitive aircraft leasing companies [5]. At the same time, the plan will also promote China to become an important gathering place for the global aircraft leasing industry.

In the past two years, the Boeing 737 MAX crisis and especially COVID-19 has had a systemic impact on the aviation industry, with the latter bringing the aviation industry to a halt, resulting in billions of dollars in losses around the world [6]. The aircraft leasing industry is closely related to the aviation industry. COVID-19 has led to a significant reduction in air travel and the grounding of aircraft fleets. Most airlines have struggled to generate cash and have been unable to make payments for leased aircraft. It has led to these companies seeking to have their payments deferred until air travel demand returns to normalcy. Their orders for leasing aircraft have either been canceled or postponed [7]. Due to the long-term impact of COVID-19, the aircraft leasing industry will inevitably be adversely affected. In this context, it is necessary to study the sustainable development mechanism of the aircraft leasing industry and propose effective measures to support the development of this industry.

COVID-19 is a public health emergency because of its rapid spread and wide infection range, resulting in it being the most difficult disease to prevent and control in the past century. The COVID-19 pandemic has triggered a wave of economic downturns across the globe due to disruptions in supply and demand chains in the travel market. Its impact on the aviation industry, aircraft leasing industry, and aircraft production and supply capacity is unprecedented. The sustainable development of the aircraft leasing industry is a complex issue. In the research process, many factors are involved and a large amount of data needs to be collected. Currently, in the research on the aircraft leasing industry, the influence factors of the pandemic are rarely considered. At the same time, due to the lack of relevant data, discussions were generally only focused on a specific issue in the aircraft leasing industry. For example, using comparative analysis methods to study the tax issues affecting the development of the aircraft leasing industry [8], using the Stackelberg game theory method to analyze rental pricing and lease contract design problems in the aircraft leasing industry [9], and using mathematical programming models to study how airlines make decisions about aircraft leasing business [10]. These studies did not systematically analyze the relationship between various factors affecting the sustainable development of the aircraft leasing industry. There is a lack of sensitivity comparative analysis among various factors that affect the sustainable development of the aircraft leasing industry.

In this paper, we focus on the application of SD methods to the sustainable development of the aircraft leasing industry, on which current research is still not thorough enough. We first build a system dynamics model for the sustainable development of the aircraft leasing industry that considers the complicated cause-and-effect relationships and feedback mechanisms among different elements. Then VENSIM (Ventana Simulation) simulations are made to predict the lease volume and market share of Chinese aircraft leasing companies in the future under different conditions. We find that among the factors affecting the sustainable development of China's aircraft leasing industry, the change in GDP growth has a significant impact on the growth of the industry. Reducing the average financing cost and the income tax rate of aircraft leasing companies, increasing the investment in talent, and controlling risk will increase the market share of China's aircraft leasing companies, while increasing the number of companies will have little effect on market share.

The rest of this paper is as follows: Section 2 provides a relevant literature review. Section 3 presents the model framework. Section 4 describes the study area and data sources. Section 5 shows the simulation results. Section 6 discusses the key conclusions, limitations, and directions for future work. Section 7 contains concluding and policy recommendations.

## 2. Literature Review

The aircraft leasing industry has a global history of more than 60 years. Aircraft leasing is a method of fleet acquisition and has become an increasingly important tool for airline financing. Leasing has always been an important source of finance to carriers in the U.S. airline industry [11]. The aircraft leasing market serves a valuable social function by improving the allocative efficiency of airlines [12]. Research on the aircraft leasing industry in developed countries mainly focuses on the aircraft leasing market, aircraft leasing business practices, aircraft leasing decision-making, aircraft value analysis, the impact of changes in laws, taxes on the aircraft leasing business, etc. As the leasing market allows airlines the opportunity to adjust capacity, it may change their aggregate demand for aircraft. Therefore, some scholars have tried to construct the models to help airlines plan their strategic fleet acquisitions and disposals [13]. In addition, some of them demonstrated the switch in the characteristics of aircraft leasing and quantified its effects on air carrier debt burdens, using public data from 73 airlines operating around the world from 1996 to 2011 to analyze the impact of aircraft leasing options on airlines [14]. Aircraft leasing leads to use of the aircraft by the lessee, and may involve large financial institutions and investors who actually purchase the aircraft and then lease it to the party desiring use [15]. In the leasing process, local laws, taxes, policies, and other factors are involved. To control the risk of leasing business and reduce the tax impact, some scholars have analyzed the impact of aircraft leasing on international air law treaties [16] and studied the various tax implications of aircraft leasing transactions in which parties are involved [17]. Moreover, some scholars studied the problem of rental rate pricing and rental contract designing in the aircraft leasing industry [18].

On 23 September 1980, China Civil Aviation signed a B747SP aircraft lease with Hanover Leasing Company of the United States. This transaction opened the history of aircraft leasing in China. Since then, with the reform of the civil aviation industry, airlines have used aircraft leasing as one of the main channels for fleet growth to meet market demand and respond to market competition. Furthermore, Chinese scholars and industry practitioners have also begun to research the aircraft leasing industry. At present, their research includes the development prospects, development models, financing innovation, risk management, law, accounting, and tax policies of the aircraft leasing industry. In terms of the development model, they compared the aircraft leasing development experience in advanced regions abroad with that at home, compared the traditional aircraft leasing model with the SPV model [19], sorted out the problems existing, and put forward suggestions for the development of the aircraft leasing industry [20]. They analyzed the profit model of the aircraft leasing industry and elaborated on the development trend of aircraft leasing [21]. In terms of evaluating aircraft leasing companies, they used the CAMELS evaluation system to conduct empirical analysis and summarized the main gaps in China's aircraft leasing companies, including capital adequacy ratio, management level, and liquidity management, after which they put forward development suggestions for aircraft leasing companies [22]. In terms of the internal operation management of aircraft leasing companies, they used the ARMA (autoregressive moving average) model to explain the method for determining the aircraft leasing interest rate and constructed an aircraft leasing pricing model under interest rate adjustments [23]. In terms of financing, they studied aircraft leasing ABS (Asset Backed Securitization), which can expand the aircraft leasing companies' financing channels and optimize balance sheets. They believe that ABS will be an important financing channel for aircraft leasing companies in the future [24]. They also analyzed the mode of developing ABS business in the domestic bonded zone and provided a new financing method for aircraft leasing companies and domestic airlines [25]. In terms of promoting the optimization of the development environment, with the emergence and prosperity of the domestic leasing industry, new transaction structures and models of aircraft leasing have emerged, which leaves new challenges for the current legal system [26], This introduced the significance of applicable law for aircraft leasing contracts and existing practices, and the selection of applicable law for Chinese companies in aircraft leasing contracts [27]. They

suggested that the government formulate a leasing law and strengthen both financial and tax support to promote the development of the aircraft leasing industry [28].

System dynamics (SD) theory was founded in 1956 by Jay W. Forrester. It imagines the motion of all systems in the world as fluid, using a causal loop diagram, stock, and flow diagram to represent the structure of the system [29]. It has proved to be a useful methodology to support both the understanding and learning processes of complex systems and phenomena [30]. It uses computer software to make simulations to reflect the dynamic causal relationships and intrinsic mechanisms within a complicated system, which is high-order, nonlinear, and provides multi-feedback [31]. Through qualitative and quantitative methods, SD has been successfully applied to many different fields of studies (e.g., climate change, physics, engineering, environmental sciences, economics, management, etc.). In the field of transportation, some scholars have used SD to assess airport terminal performance [32,33], to evaluate the impact of future demand on the runway capacity of the airport [34], to analyze airlines' aging aircraft cost and develop a practical policy for maintenance cost reduction, to holistically represent and critically assess the different facets of MRO operations, and to help airlines analyze various decision scenarios [35]. Some scholars have used SD to study the sustainable development of civil aviation. They have studied the sustainability of the growth of the commercial aviation industry and its impact on the environment [36], analyzed regional airports, and proposed the best strategy to maintain the regional airport ecosystem [37].

In summary, scholars and industry practitioners have analyzed the relevant factors that affect the sustainable development of the aircraft leasing industry. Among these analyses, qualitative analysis is more frequently used, while quantitative analysis is relatively insufficient. In particular, few scholars use the system dynamics method to simulate and analyze the aircraft leasing industry. Therefore, this paper constructs a system dynamics model for the sustainable development of China's aircraft leasing industry, simulates the main factors that affect the industry's development, and proposes policy suggestions for its development.

## 3. The Framework Model

From a system perspective, among the core elements of the aircraft leasing industry, aircraft leasing products, supply entities, demand entities, and the market itself constitute the interior of the sustainable development system of the aircraft leasing industry. Aircraft manufacturers, leasing supervision bodies, and leasing environments constitute the exterior of the sustainable development system of the aircraft leasing industry. According to the relationship between the elements, the sustainable development system of the aircraft leasing industry can be divided into the economic, market, enterprise, and environmental subsystems. The four subsystems influence and interact with each other.

The economic subsystem mainly reflects the demand for aircraft leasing, with factors including China's GDP (gross domestic product), total air transport turnover, total financial assets, capital demand and supply, aircraft demand, etc. The market subsystem mainly reflects the supply of aircraft leasing, with factors including the number of aircraft leasing companies in China, aircraft leasing market risk, profitability, market share, etc. The enterprise subsystem is mainly within the main body of aircraft leasing supply, including professional talent, management capabilities, innovation capabilities, profitability, market share, etc. The environmental subsystem mainly reflects aircraft manufacturers, aircraft leasing market supervision entities, and the development environment of the aircraft leasing industry, including aircraft supply capabilities, laws, accounting, taxation, regulatory policies, etc.

### 3.1. Model Causality Diagram and Main Feedback Loops

A causal loop diagram is an important tool for representing and visualizing the feedback structure of a system. Causal loop diagrams contain a number of feedback loops and variables, with these variables connected by arrows to reflect the casual relationships.

Each feedback loop has its own polarity, with some "+", and others "−", showing how the relative variables will change when one variable changes. Ceteris paribus, the sustainable development system model of China's aircraft leasing industry is constructed based on the four subsystems, with the causality diagram shown in Figure 1.

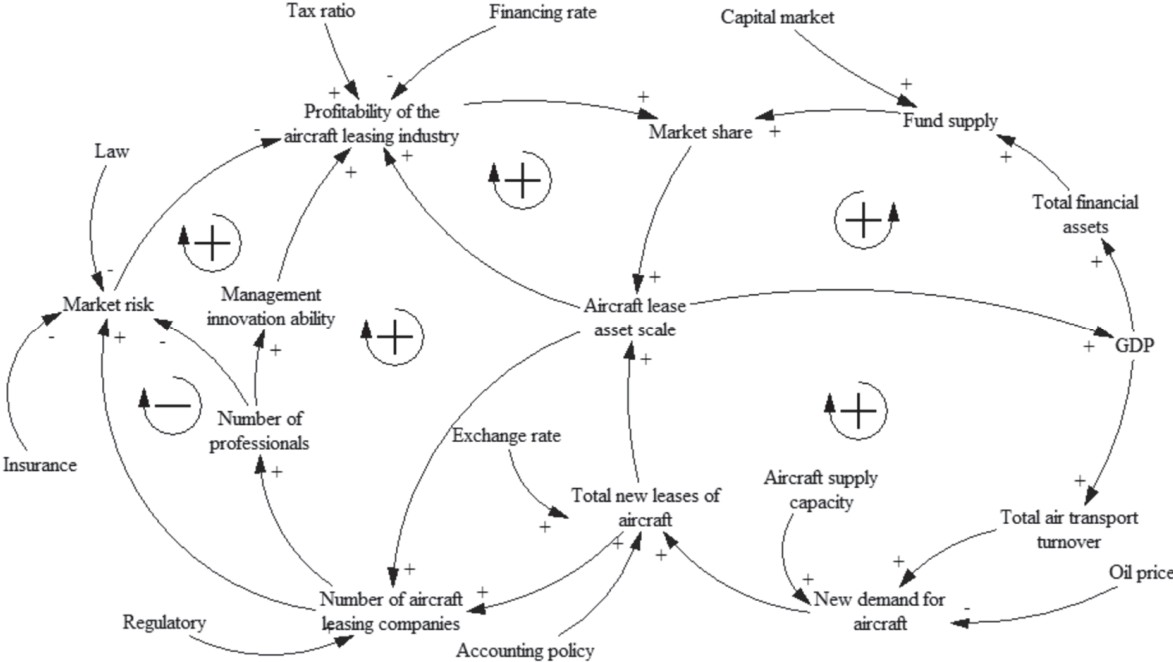

**Figure 1.** Causal loop diagram of the sustainable development system.

The main feedback loops in the causal loop diagram of the sustainable development system are as follows:

Loop 1:  GDP →+Total air transport turnover →+New demand for aircraft →+Total new leases of aircraft →+Aircraft lease asset scale →+GDP

Loop 2:  GDP →+Total financial assets →+Fund supply →+Market share →+Aircraft lease asset scale →+GDP

Loop 3:  Number of aircraft leasing companies →+Market risk →-Profitability of the aircraft leasing industry →+Market share →+Aircraft lease asset scale →+Number of aircraft leasing companies

Loop 4:  Number of aircraft leasing companies →+Number of professionals → +Management innovation ability →+Profitability of the aircraft leasing industry → +Market share →+Aircraft lease asset scale →+Number of aircraft leasing companies

Loop 5:  Number of aircraft leasing companies →+Number of professionals →-Market risk →-Profitability of the aircraft leasing industry →+Market share →+Aircraft lease asset scale →+Number of aircraft leasing companies

Loop 6:  Aircraft lease asset scale →+Profitability of the aircraft leasing industry →+Market share →+Aircraft lease asset scale

Among the six causal loops, loops 1, 2, 4, 5, and 6 are positive causal loops and loop 3 is a negative causal loop. Loop 1 includes five variables, which indicates: if GDP increases, the total air transport turnover will increase, which will increase the demand for air transport and the total number of aircraft leased by airlines will increase. Correspondingly, the scale of aircraft leasing assets of aircraft leasing companies will also increase. Loop 2 indicates: if GDP increases, financial assets will increase and the amount of funds obtained by aircraft leasing companies from the market will also increase. It will push aircraft leasing companies to increase their market share, and the scale of aircraft leasing assets of aircraft leasing companies will increase accordingly. Loop 3 indicates: if the number of aircraft leasing companies increases, it will intensify market competition, increase market risk, reduce

the profits of aircraft leasing companies, and affect the market share of aircraft leasing companies and the scale of leased assets. Loop 4 indicates: if the number of aircraft leasing companies increases, the number of aircraft professionals will increase, the management innovation capabilities of aircraft leasing companies will increase, and the market share, profits, and the scale of leased assets will also increase. Loop 5 indicates: if the number of aircraft leasing companies increases, the number of aircraft professionals will increase and the market risk of aircraft leasing companies will be better controlled, which will correspondingly affect the market share, profits, and the scale of leased assets. Loop 6 indicates: if the leased assets increase, the profit and market share of the aircraft leasing companies will increase, which will push it to increase its asset scale.

### 3.2. System Flow Diagram

The causality diagram describes the basic relationships of the system elements' causal feedback. The system flow diagram not only shows the cause-and-effect relationship among system constitute factors, but also clarifies the variables' internal natures and the scale of change in the system. Based on the causality diagram of the development system of the aircraft leasing industry, this paper uses the system dynamics method to carry out a quantitative analysis and constructs the development system flow diagram of the aircraft leasing industry, as shown in Figure 2.

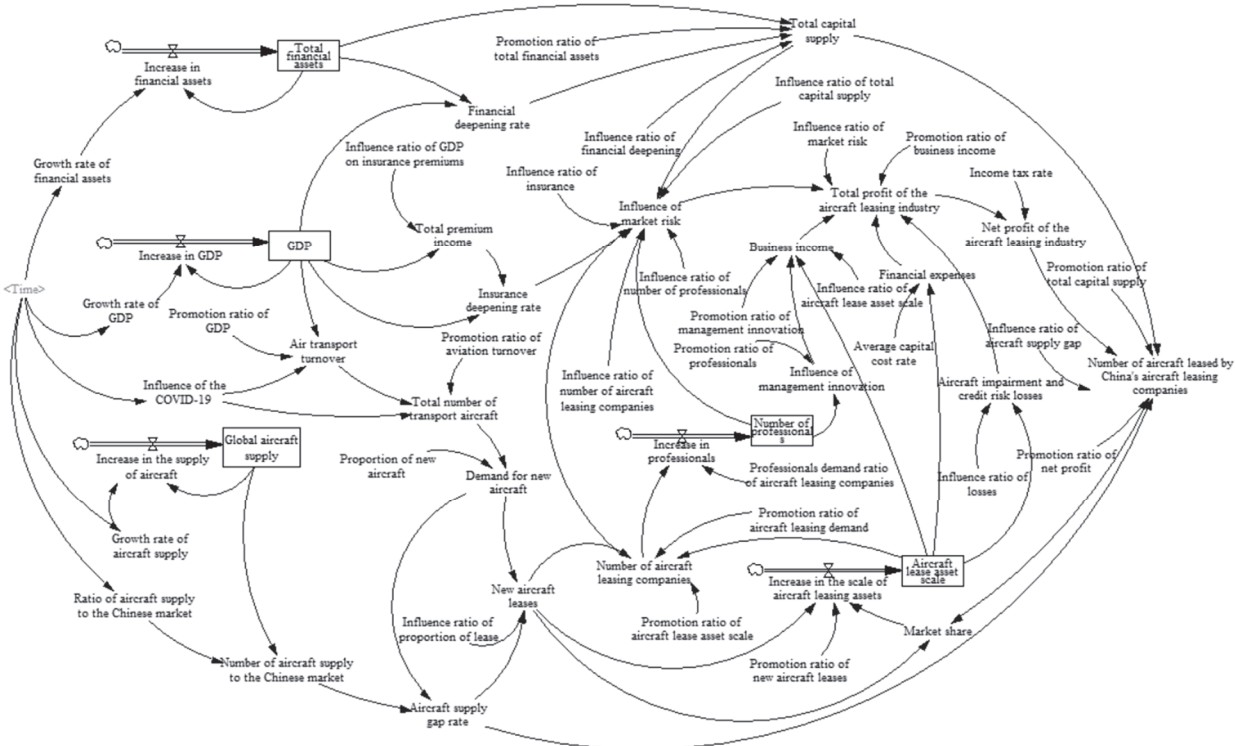

**Figure 2.** Stock-flow diagram of the sustainable development system.

In Figure 2, the variables in the box are stock variables, the variables under the arrows are rate variables, and the others are auxiliary variables and constants. The value of each stock variable, which is an accumulated value over time, is the difference between their inflow volume and outflow volume. In the diagram, five variables are represented as stock variables, namely "Total financial assets", "GDP", "Global aircraft supply", "Number of professionals", and "Aircraft lease asset scale". Flow variables include the inflow and outflow variables, which reflect the change rate of the stock variables' values. In this diagram, five variables are represented as flow variables, namely "Increase in financial assets", "Increase in GDP", "Increase in the supply of aircraft", "Increase in professionals", and "Increase in the scale of aircraft leasing assets". Auxiliary variables are the intermediate

variables between the stock variables and the flow variables. In the diagram, there are a number of auxiliary variables which are indispensable in detailing and describing the intrinsic mechanism within the system. For example, the auxiliary variable "Number of aircraft leasing companies" is directly related to "Aircraft lease asset scale", which is a stock variable, and it will also influence "Increase in professionals", which is a flow variable. The constants are system parameters with invariable values, such as "Promotion ratio of GDP", "Promotion ratio of aviation turnover", "Promotion ratio of aircraft lease asset scale", "Promotion ratio of professionals", etc. We need to determine the values of the constants based on related materials and field investigation, then input them into the model. Moreover, we should establish the mathematical equations to quantitatively describe the relationships among different variables.

*3.3. The Model Equation*

System equations are a set of mathematical relations that quantitatively describe the relationships between system elements. In system dynamics, the stock, flow, auxiliary variables, and constants are typically used to construct the system equation based on the system flow diagram. Taking the stock GDP in Figure 2 as an example, the system equation is constructed as follows:

$$GDP.k = GDP.J + GDP\ increment.K.J \times DT \tag{1}$$

$$GDP\ increment.K.J = GDP.J \times GDP\ rate \tag{2}$$

Among them, *GDP.K* represents the value of stock GDP at time K, *GDP.J* represents the value of stock *GDP* at time J, *GDP increment.K.J* represents the flow of *GDP* in the KJ time interval, and *GDP rate* is the growth rate of GDP. *GDP increment.K.J* is calculated from *GDP* at time J according to the growth rate of GDP. The equation for each stock is an integral equation. In this paper, we use VENSIM software to build the system dynamics model. To simplify the expression, the INTEG integral function is introduced into the above equation. Its functional form is INTEG(x, y), with x being the amount of variable growth and y being the initial value. According to the development system flow diagram of the aircraft leasing industry, the equations between the various variables are established. Considering that the number of aircraft, the number of talents, and the number of companies are all integers, the INTEGER rounding function needs to be used, with the function expression being INTEGER(x).

1.  Stock variables

$$GDP = INTEG\ (Increase\ in\ GDP,\ Initial\ value) \tag{3}$$

$$Total\ financial\ assets = INTEG\ (Increase\ in\ financial\ assets,\ Initial\ value) \tag{4}$$

$$Global\ aircraft\ supply = INTEG\ (Increase\ in\ the\ supply\ of\ aircraft,\ Initial\ value) \tag{5}$$

$$Aircraft\ lease\ asset\ scale = INTEG\ (Increase\ in\ the\ scale\ of\ aircraft\ leasing\ assets,\ Initial\ value) \tag{6}$$

$$Number\ of\ professionals = INTEG\ (Increase\ in\ professionals,\ Initial\ value) \tag{7}$$

2.  Rate variables

$$Increase\ in\ GDP = GDP \times Growth\ rate\ of\ GDP \tag{8}$$

$$Increase\ in\ financial\ assets = Total\ financial\ assets \times Growth\ rate\ of\ financial\ assets \tag{9}$$

$$Increase\ in\ the\ supply\ of\ aircraft = INTEGER\ (Global\ aircraft\ supply \times Growth\ rate\ of\ aircraft\ supply) \tag{10}$$

$$Increase\ in\ the\ scale\ of\ aircraft\ leasing\ assets = New\ aircraft\ leases \times Market\ share \times Promotion\ ratio\ of\ new\ aircraft\ leases \tag{11}$$

$$Increase\ in\ professionals = INTEGER\ (Number\ of\ aircraft\ leasing\ companies \times Professionals\ demand\ ratio\ of\ aircraft\ leasing\ companies) \tag{12}$$

3.  Auxiliary variables

$$\text{Total premium income} = \text{GDP} \times \text{Influence ratio of GDP on insurance premiums} \tag{13}$$

$$\text{Insurance deepening rate} = \text{Total premium income}/\text{GDP} \tag{14}$$

$$\text{Financial deepening rate} = \text{Total financial assets}/\text{GDP} \tag{15}$$

$$\text{Total capital supply} = \text{Total financial assets} \times \text{Promotion ratio of total financial assets} + \text{Financial deepening rate} \times \text{Influence ratio of financial deepening} \tag{16}$$

$$\text{Air transport turnover} = \text{GDP} \times \text{Promotion ratio of GDP} \times \text{Influence of COVID-19} \tag{17}$$

$$\text{Total number of transport aircraft} = \text{INTEGER (Air transport turnover}/\text{Promotion ratio of aviation turnover}/\text{Influence of COVID-19)} \tag{18}$$

$$\text{Demand for new aircraft} = \text{INTEGER (Total number of transport aircraft} \times \text{Proportion of new aircraft)} \tag{19}$$

$$\text{Number of aircraft supply to the Chinese market} = \text{INTEGER (Global aircraft supply} \times \text{Ratio of aircraft supply to the Chinese market)} \tag{20}$$

$$\text{Ratio of aircraft supply to the Chinese market} = \text{(Demand for new aircraft} - \text{Number of aircraft supply to the Chinese market)}/\text{Demand for new aircraft} \tag{21}$$

$$\text{New aircraft leases} = \text{INTEGER (Proportion of lease} \times \text{(Demand for new aircraft} \times (1 - \text{Ratio of aircraft supply to the Chinese market)))} \tag{22}$$

$$\text{Number of aircraft leasing companies} = \text{INTEGER (New aircraft leases} \times \text{Promotion ratio of aircraft leasing demand} + \text{Aircraft lease asset scale} \times \text{Promotion ratio of aircraft lease asset scale)} \tag{23}$$

$$\text{Influence of management innovation} = \text{Number of professionals} \times \text{Promotion ratio of professionals} \tag{24}$$

$$\text{Influence of market risk} = \text{Number of aircraft leasing companies} \times \text{Influence ratio of number of aircraft leasing companies} + \text{Number of professionals} \times \text{Influence ratio of number of professionals} + \text{Insurance deepening rate} \times \text{Influence ratio of insurance} + \text{Total capital supply} \times \text{Influence ratio of total capital supply} \tag{25}$$

$$\text{Business income} = \text{Influence of management innovation} \times \text{Promotion ratio of management innovation} + \text{Aircraft lease asset scale} \times \text{Promotion ratio of aircraft lease asset scale} \tag{26}$$

$$\text{Financial expenses} = \text{Aircraft lease asset scale} \times \text{Average capital cost rate} \tag{27}$$

$$\text{Aircraft impairment and credit risk losses} = \text{Aircraft lease asset scale} \times \text{Influence ratio of losses} \tag{28}$$

$$\text{Total profit of the aircraft leasing industry} = \text{(Business income} - \text{Financial expenses)} \times \text{Promotion ratio of business income} + \text{(Influence of market risk} - \text{Aircraft impairment and credit risk losses)} \times \text{Influence ratio of market risk} \tag{29}$$

$$\text{Net profit of the aircraft leasing industry} = \text{Total profit of the aircraft leasing industry} \times \text{Income tax rate} \tag{30}$$

$$\text{Number of aircraft leased by China's aircraft leasing companies} = \text{IF THEN ELSE(Ratio of aircraft supply to the Chinese market} = 0, \text{(INTEGER (Net profit of the aircraft leasing industry} \times \text{Promotion ratio of net profit} + \text{Total capital supply} \times \text{Promotion ratio of total capital supply)), (INTEGER ((Net profit of the aircraft leasing industry} \times \text{Promotion ratio of net profit} + \text{Total capital supply} \times \text{Promotion ratio of total capital supply)} \times (1 - \text{Ratio of aircraft supply to the Chinese market} \times \text{Influence ratio of aircraft supply gap)))} \tag{31}$$

$$\text{Market share} = \text{Number of aircraft leased by China's aircraft leasing companies}/\text{New aircraft leases} \tag{32}$$

## 4. Parameter Determination and Model Validation

### 4.1. Parameter Determination

Parameter determination is the key to the system dynamics model. In general, there are four methods for determining the parameters of the system dynamics model. One is to take surveys to determine parameters, listen to industry experts' practice summary opinions, or adopt trend analysis based on actual data. The second is to determine the parameters through questionnaire surveys, set up questionnaires for key indicators, and invite professionals to complete them. After screening and generating statistics, the parameter values of the relevant indicators are summarized and calculated. The third is to determine parameters through econometric methods such as regression. The fourth is to determine parameters based on input and output. This method is generally used to study the influence coefficients between related industries.

Based on China's civil aviation bulletin, financial statistics, leasing industry statistics, etc., the relevant data involved in the sustainable development system model of China's aircraft leasing industry is shown in Tables 1–3. The time horizon is from 2007 to 2016, representing 10 years of data.

**Table 1.** Data of the main indicators of China's aviation industry and financial industry from 2007 to 2016.

| Year | Total Number of Transport Aircraft | Annual Increase | Air Transport Turnover (Billion Tons) | GDP (Billion Yuan) | M2 (Billion Yuan) |
|------|-----------------------------------|-----------------|---------------------------------------|--------------------|--------------------|
| 2007 | 1134 | 136 | 36.53 | 27,009.23 | 40,340.13 |
| 2008 | 1259 | 125 | 37.66 | 31,924.46 | 47,516.66 |
| 2009 | 1417 | 158 | 42.71 | 34,851.77 | 61,022.45 |
| 2010 | 1597 | 180 | 53.845 | 41,211.93 | 72,585.18 |
| 2011 | 1764 | 167 | 57.744 | 48,794.02 | 85,159.09 |
| 2012 | 1941 | 177 | 61.032 | 53,858.00 | 97,414.88 |
| 2013 | 2145 | 204 | 67.172 | 59,296.32 | 110,652.50 |
| 2014 | 2370 | 225 | 74.812 | 64,356.31 | 122,837.48 |
| 2015 | 2650 | 280 | 85.165 | 68,885.82 | 139,227.81 |
| 2016 | 2950 | 300 | 96.251 | 74,639.51 | 155,006.67 |

**Table 2.** Data of the main indicators of China's aircraft leasing industry from 2007 to 2016.

| Year | Number of Aircraft Leasing Companies | Number of Professionals | Aircraft Lease Asset Scale (Billion) | New Lease Amount (Billion) | Total Risk Assets (Billion) | Operating Lease Assets (Billion) | Income (Billion) | Financial Expense (Billion) |
|------|--------------------------------------|-------------------------|--------------------------------------|----------------------------|-----------------------------|----------------------------------|------------------|-----------------------------|
| 2007 | 12 | 138 | 6.75 | 8.63 | 0.00 | 7.98 | 0.86 | 0.32 |
| 2008 | 12 | 179 | 16.05 | 12.05 | 0.12 | 11.54 | 1.29 | 0.50 |
| 2009 | 12 | 216 | 32.02 | 20.36 | 0.25 | 16.70 | 2.33 | 0.87 |
| 2010 | 17 | 321 | 63.14 | 40.97 | 0.18 | 31.31 | 4.77 | 2.12 |
| 2011 | 17 | 376 | 105.36 | 58.16 | 0.31 | 41.26 | 9.91 | 5.33 |
| 2012 | 20 | 470 | 159.73 | 82.96 | 0.44 | 58.26 | 16.08 | 9.27 |
| 2013 | 22 | 589 | 202.51 | 88.45 | 0.74 | 69.20 | 20.74 | 11.40 |
| 2014 | 26 | 686 | 255.37 | 109.72 | 1.26 | 106.27 | 25.20 | 15.53 |
| 2015 | 40 | 939 | 326.29 | 142.17 | 1.77 | 145.13 | 28.16 | 16.25 |
| 2016 | 52 | 1109 | 397.91 | 174.67 | 2.15 | 225.14 | 31.66 | 16.60 |

According to the above data, the initial values of total financial assets, GDP, global aircraft production and supply, number of professionals, and aircraft leasing assets are RMB 40,340.13 billion, RMB 27,009.23 billion, 909 aircraft, 138 people, and RMB 6.746 billion, respectively. The use of model parameter determination methods, through regression analysis, expert estimation, etc., determines the values of the variables in the model, as shown in Table 4.

**Table 3.** Data of Boeing and Airbus aircraft deliveries to China from 2007 to 2016.

| Year | Number of Boeing Aircraft Delivered | Number of Airbus Delivered Aircraft | Total Number of Aircraft Delivered to China | Number of Aircraft Delivered by Leasing Company | Number of Aircraft Delivered by Chinese Leasing Companies |
|---|---|---|---|---|---|
| 2007 | 456 | 453 | 158 | 112 | 8 |
| 2008 | 391 | 483 | 134 | 92 | 20 |
| 2009 | 462 | 471 | 169 | 58 | 8 |
| 2010 | 457 | 505 | 200 | 98 | 23 |
| 2011 | 470 | 531 | 201 | 120 | 39 |
| 2012 | 592 | 585 | 247 | 127 | 42 |
| 2013 | 639 | 621 | 303 | 170 | 71 |
| 2014 | 708 | 626 | 339 | 238 | 124 |
| 2015 | 747 | 632 | 374 | 236 | 117 |
| 2016 | 729 | 693 | 360 | 251 | 139 |

**Table 4.** Related parameter values in the sustainable development system model of China's aircraft leasing.

| Name of Variables | Value |
|---|---|
| Growth rate of GDP | 0.11910 |
| Promotion ratio of GDP | 0.00126 |
| Influence ratio of GDP on insurance premiums | 0.05100 |
| Growth rate of aircraft supply | 0.05245 |
| Promotion ratio of aviation turnover | 0.31868 |
| Proportion of new aircraft | 0.12800 |
| Influence ratio of proportion of lease | 0.78020 |
| Professionals demand ratio of aircraft leasing companies | 4.22490 |
| Promotion ratio of aircraft lease asset scale | 0.00878 |
| Promotion ratio of aircraft leasing demand | 0.02299 |
| Influence ratio of number of aircraft leasing companies | −0.07404 |
| Influence ratio of number of professionals | −0.04335 |
| Influence ratio of insurance | 61.04902 |
| Promotion ratio of professionals | 2.17410 |
| Promotion ratio of management innovation | 0.01950 |
| Influence ratio of aircraft lease asset scale | 0.09480 |
| Promotion ratio of new aircraft leases | 4.25350 |
| Promotion ratio of business income | 0.54710 |
| Influence ratio of market risk | −1.08760 |
| Influence ratio of total capital supply | 0.01360 |
| Promotion ratio of total capital supply | 0.06690 |
| Promotion ratio of net profit | 0.28500 |
| Income tax rate | 0.25 |
| Influence ratio of losses | 0.00100 |
| Average capital cost rate | 0.03200 |
| Influence ratio of aircraft supply gap | 1.15000 |

COVID-19 will continue to affect the aviation industry in the coming years. According to a forecast provided by the International Air Transport Association, air passenger traffic will return to the pre-COVID-19 level in 2024. The Civil Aviation Administration of China proposed in 2021 that aviation turnover would return to more than 80% of what it was before COVID-19. Based on the above two points, from 2020 to 2024, the impact of COVID-19 on the total decrease in demand for transport aircraft will gradually recover from 40% to zero.

In 2019 and 2020, China's economy was affected by the trade war and COVID-19, with the GDP growth rate decreasing significantly. From 2021 to 2025, China's five-year development plan proposes an average annual GDP growth rate of about 6%. From 2007 to 2016, variables such as the growth rate of financial assets and the proportion of aircraft supply in the Chinese market were not constant values. Through regression analysis, this

paper establishes the trend function of the growth rate of financial assets and the proportion of aircraft supply in the Chinese market, as follows:

$$\text{The growth rate of financial assets} = -0.06 \times \text{LN (Time} - 2006) + 0.2437$$

$$\text{The proportion of aircraft supply in the Chinese market} = 0.0107 \times (\text{Time} - 2007) + 0.1652$$

Among them, Time is the annual value of model simulation.

### 4.2. Model Validation

Model validation is necessary to check whether the constructed system dynamics model is consistent with the real system. This paper adopts two methods, running test and historical test, to assess the validity of the model.

To examine the stability of the model, the simulation step length DT is modified to compare and analyze the running results. The following values were chosen for simulation: DT = 0.25, DT = 0.5, and DT = 1. The simulation results of the model's main variables "market share" and "new aircraft leases" are shown in Figure 3. The results show that the behavior of the system is stable.

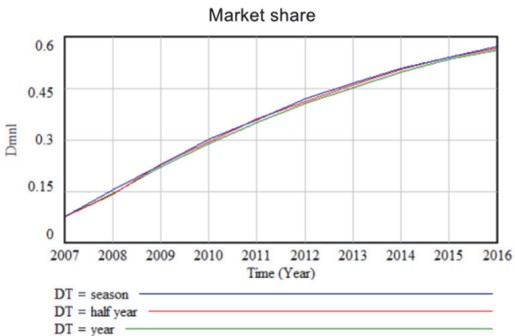
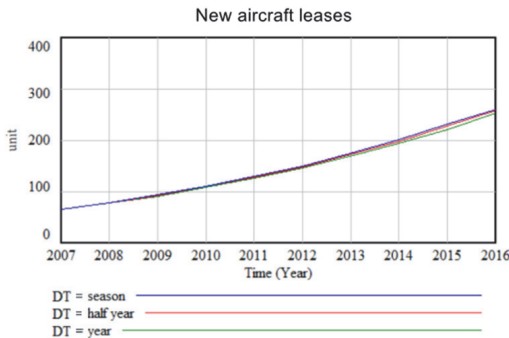

**Figure 3.** Simulation results for market share and new aircraft leases for different values of DT.

In order to check whether the simulation results of the model were consistent with the real system, simulations were started in 2007 to observe the market share, the number of newly leased aircraft, and the global aircraft supply in 2016, 2017, and 2018. The simulation results are shown in Table 5. The simulation results show that compared with the historical data of the actual system, the absolute values of the relative errors of the main variables are all less than 5%. With the exception of the market share error of 3.49% in 2016, the remaining relative errors are all less than 3%. The simulated values were fitted with the actual values. This shows that the constructed model can simulate the growth system of China's aircraft leasing industry, is effective, and has high credibility.

**Table 5.** Historical test of the sustainable development model of China's aircraft leasing industry.

| Name of Variables | 2016 | | | 2017 | | | 2018 | | |
|---|---|---|---|---|---|---|---|---|---|
| | True Value | Simulated Value | Error | True Value | Simulated Value | Error | True Value | Simulated Value | Error |
| Market share | 0.5538 | 0.5731 | 3.49% | 0.6090 | 0.5951 | −2.28% | 0.6101 | 0.6127 | 0.43% |
| Number of newly leased aircraft | 251 | 253 | 0.80% | 289 | 284 | −1.73% | 318 | 315 | −0.94% |
| Global aircraft supply | 1422 | 1434 | 0.84% | 1474 | 1509 | 2.37% | 1585 | 1588 | 0.19% |

## 5. Simulation Prediction and Sensitivity Analysis

### 5.1. Simulation Result Output

VENSIM software is used to simulate the dynamic model of the growth system of China's aircraft leasing industry. The simulation run time is from 2007 to 2025 and the

simulation step size is one year. After the simulation operation, the output results of the lease volume and market share of the China's aircraft leasing companies are shown in Figure 4.

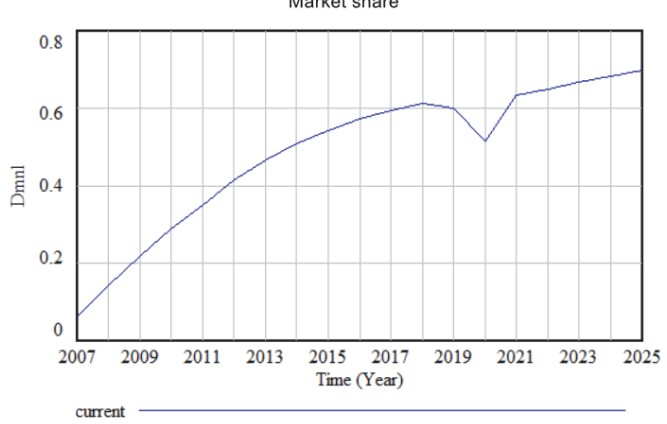 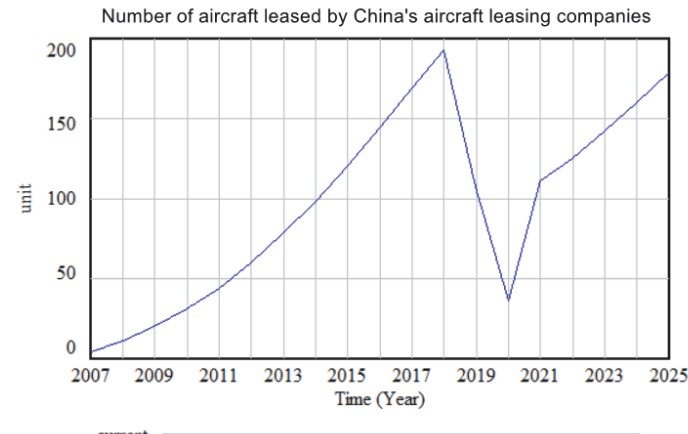

**Figure 4.** Simulated trends for market share and leasing volume of Chinese aircraft leasing companies from 2007 to 2025.

The simulation results show that the market share and leasing volume of Chinese aircraft leasing companies have fallen sharply in 2019 and 2020 due to the Boeing 737 MAX incident and COVID-19. After 2021, they are forecast to increase year by year. The lease volume in 2025 is predicted to return to the 2018 level. The Chinese aircraft leasing companies' market share declined significantly in 2020. The main reason was that the global aviation market experienced a sharp decline due to the impact of COVID-19. China actively implements prevention and control measures and has become the world's largest domestic air passenger transport market, with a global share of 19.9%. The major global aircraft leasing companies adjusted their business strategies and increased their business expansion in the Chinese market, which intensified competition in the Chinese aircraft leasing market. In this context, the market share of Chinese aircraft leasing companies has dropped by nearly 10%. However, with the recovery of China's economic growth in 2021, the market share of China's aircraft leasing companies is forecast to return to the 2018 level in 2021 and then maintain a steady growth trend. By the end of 2025, the market share will reach 69.65%, as shown in Table 6.

**Table 6.** The leasing volume and market share of China's aircraft leasing companies from 2018 to 2025.

| Year | Leasing Volume | Market Share |
|------|----------------|--------------|
| 2018 | 193 | 61.27% |
| 2019 | 107 | 60.11% |
| 2020 | 36 | 51.43% |
| 2021 | 111 | 63.43% |
| 2022 | 125 | 64.77% |
| 2023 | 142 | 66.67% |
| 2024 | 160 | 68.38% |
| 2025 | 179 | 69.65% |

*5.2. Sensitivity Analysis*

Sensitivity analysis is used to simulate the changing trend of the system by adjusting the values of the variables in the model. It enables a comparison and analysis of the degree of influence of key variables on the system, and the determination of the important variables that affect the system. This paper selects variables from the economic,

market, enterprise, and environmental subsystems to simulate and analyze their impact on the leasing volume and market share of China's aircraft leasing companies, and then analyzes the impact on the sustainable development of China's aircraft leasing industry.

### 5.2.1. Analysis of the Impact of GDP Growth Rate

We use the GDP growth rate to analyze the impact of the Chinese macro economy on the market share of domestic aircraft leasing companies. From 2021 to 2025, suppose that China's average GDP growth rate is increased by 20% and decreased by 20%, while other parameters of the system model remain unchanged. The simulated values of the lease volume and market share of Chinese aircraft leasing companies are generated through VENSIM, with the output shown in Figure 5. It can be seen that after the average GDP growth rate increases by 20%, the market share will drop from 69.65% to 64.98% in 2025, a decrease of 4.67%. After the increase in GDP growth rate, the market share of Chinese aircraft leasing companies declined. The reason is that GDP growth has led to a further increase in the demand for aircraft leasing in China, but the supply capacity of domestic aircraft leasing companies cannot meet market needs. At this time, aircraft leasing companies in other countries filled the market demand, which led to a decline in the market share of Chinese aircraft leasing companies.

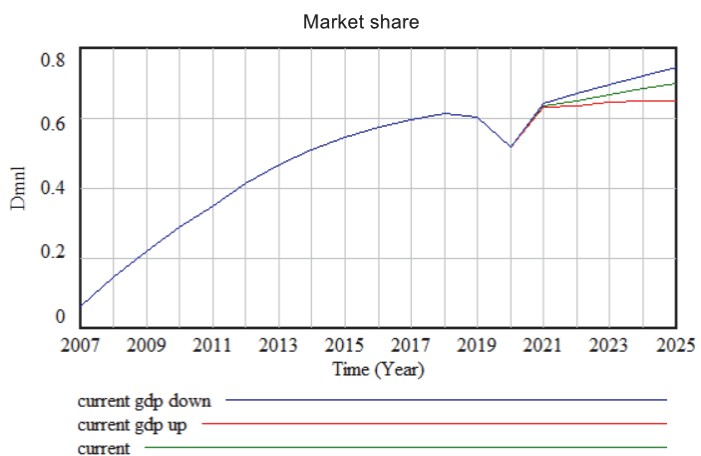 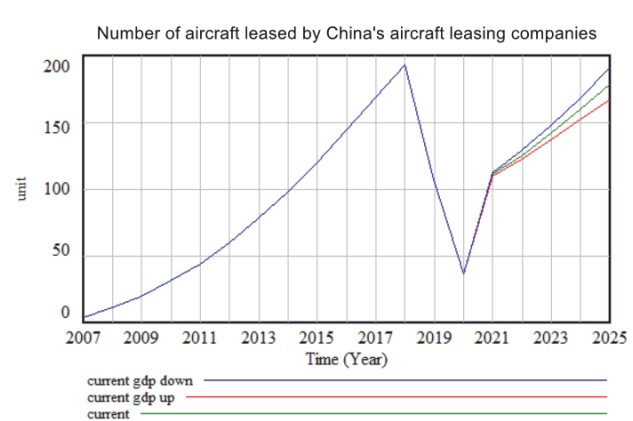

**Figure 5.** The influence of the GDP growth rate on market share and lease volume.

### 5.2.2. Analysis of the Impact of the Number of Aircraft Leasing Companies

The promotion coefficient of the scale of leasing assets will affect the growth of the number of leasing companies. We use the change in the value of this coefficient to analyze the impact of the number of leasing companies on the market share of Chinese aircraft leasing companies. From 2007 to 2025, suppose that the coefficient is increased by 20% and decreased by 20%, while the other parameters of the system model remain unchanged. The simulated values of the lease volume and market share of the Chinese aircraft leasing companies are generated through VENSIM, with the output shown in Figure 6. The simulation shows that after the increase in the number of Chinese aircraft leasing companies, their market share actually fell, from 69.65% to 69.26%, a decrease of 0.39%. This result shows that when the number of Chinese aircraft leasing companies reaches a certain level, the increase of market players will lead to intensified competition, which is not conducive to the overall development of the aircraft leasing industry.

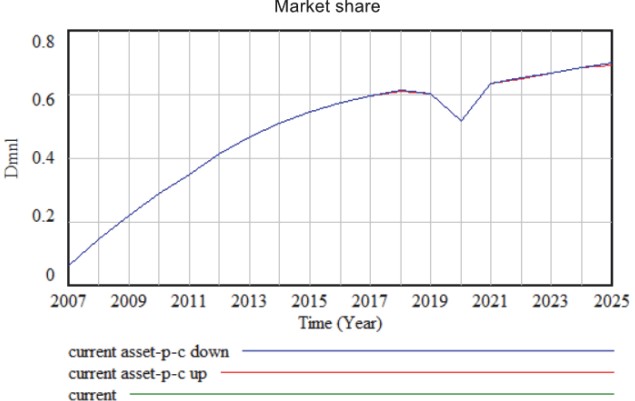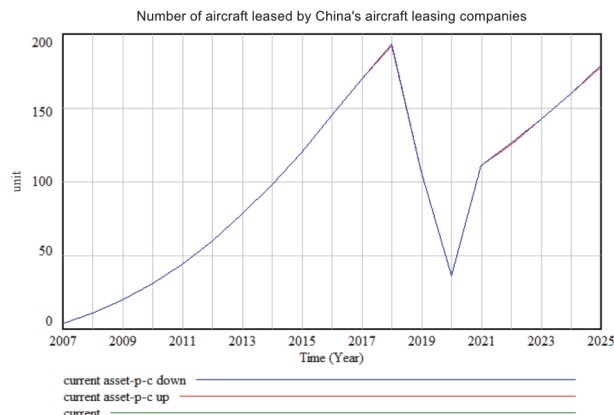

**Figure 6.** The influence of the number of aircraft leasing companies on market share and lease volume.

### 5.2.3. Analysis of the Impact of Aircraft Leasing Industry Risk

The level of risk affects the total profits of China's aircraft leasing industry. Through the changes in the impact of market risk, impairment, and credit risk losses, we analyze the influence on the market share of Chinese aircraft leasing companies. From 2007 to 2025, suppose that the market risk impact, the impairment, and credit risk losses increase by 20% and decrease by 20%, the simulated values of the lease volume and market share of the Chinese aircraft leasing companies are generated through VENSIM, with the output shown in Figure 7. The simulation shows that market share will drop from 69.65% to 68.87% in 2025 if the risk increases by 20%. On the contrary, if the risk decreases by 20%, market share will be 70.39% in 2025, an increase of about 0.74%. The simulation shows that rising risk hurts the market share. However, this increase of 20% risk fluctuation has not had a great impact on China's aircraft leasing industry, reflecting the industry's strong ability to resist risk.

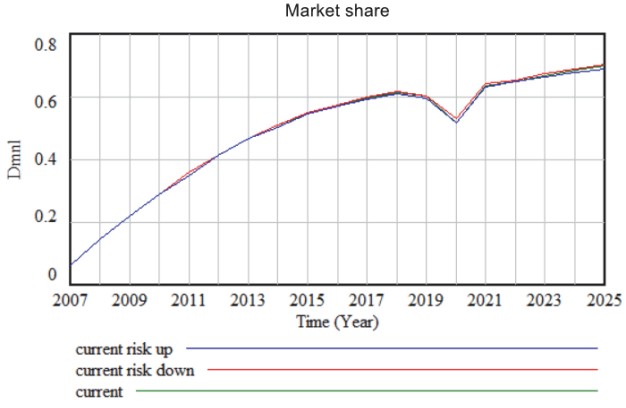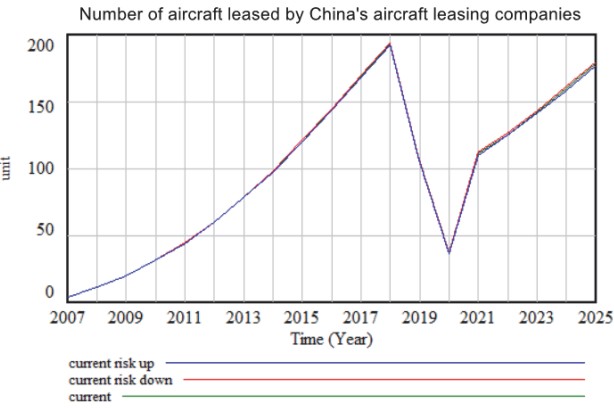

**Figure 7.** The influence of aircraft leasing companies' risk on market share and lease volume.

### 5.2.4. Analysis on the Impact of the Talent Promotion Coefficient

Aircraft leasing companies increase their investment in professional talent, which will promote the improvement of risk management and professional capabilities. It will also increase operating income. Through the change of the talent promotion coefficient, we analyze the impact of talent investment on the market share of Chinese aircraft leasing companies. Suppose that the coefficient is increased by 20% and decreased by 20%, the simulated values of the market share of the Chinese aircraft leasing companies are generated through VENSIM, with the output shown in Figure 8. The simulation shows that if there is an increase in talent investment, the market share of China's aircraft leasing companies will increase significantly. In 2025, the market share will rise from 69.65% to 70.43%. Increasing

investment in talent will have a positive impact on the sustainable development of aircraft leasing companies.

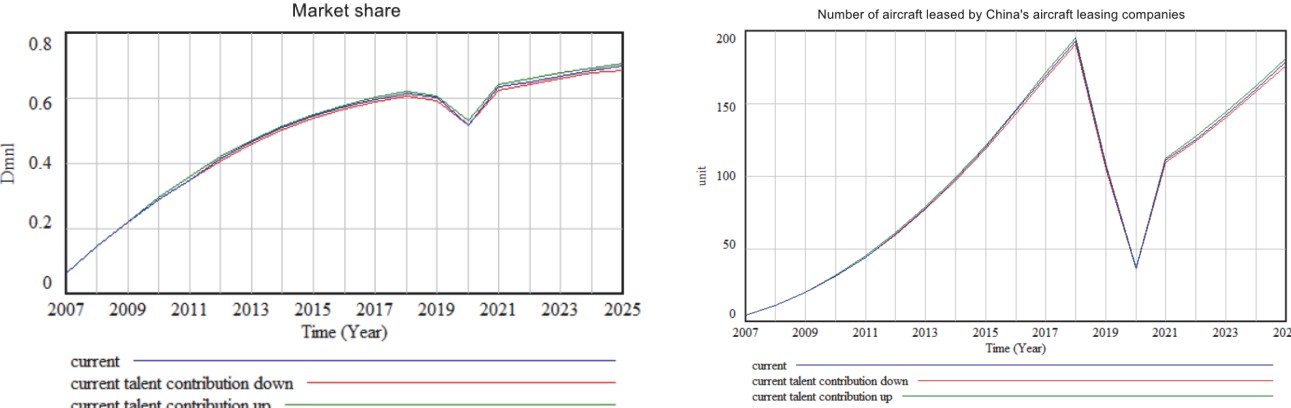

**Figure 8.** The influence of the talent promotion coefficient on market share and lease volume.

### 5.2.5. Analysis of the Impact of Income Tax Rate

Changes in income tax will affect the overall profits of the aircraft leasing industry. Through changes in income tax rates, we analyze the impact of tax policies on the market share of China's aircraft leasing companies. Suppose that the income tax rate of China's aircraft leasing industry increases by 20% and decreases by 20%, the simulated values of the lease volume and market share of China's aircraft leasing companies are generated through VENSIM, with the output shown in Figure 9. The simulation shows that if income tax drops by 20%, market share will rise from 69.65% to 70.82% in 2025. Lowering the tax rate of the aircraft leasing industry will help increase the market share of Chinese aircraft leasing companies.

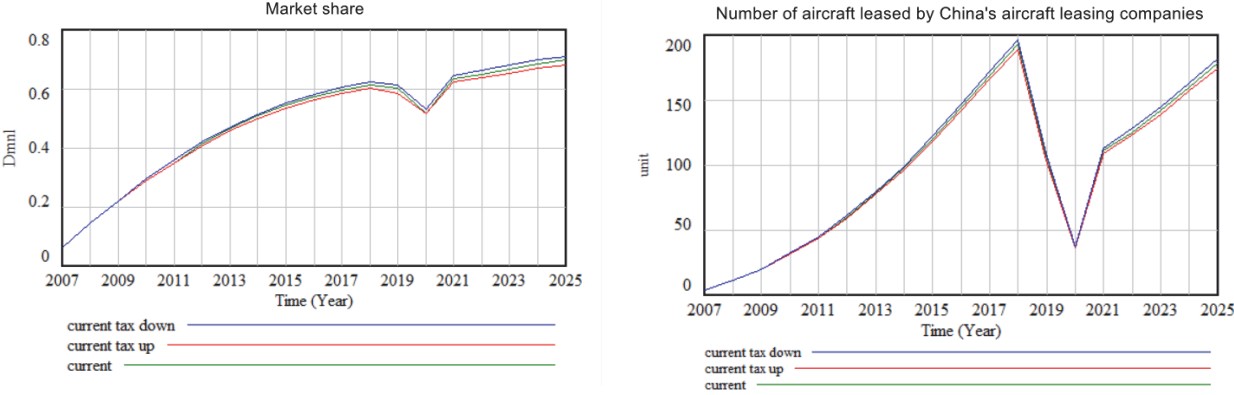

**Figure 9.** The influence of aircraft leasing companies' income tax rate on market share and lease volume.

### 5.2.6. Analysis of the Impact of the Average Financing Cost Rate

The average financing cost rate affects the operating costs of the aircraft leasing industry, thereby affecting its profit and market share. Through the change of the average financing cost rate, we analyze the impact of financing cost on the market share of Chinese aircraft leasing companies. Suppose that the average financing cost rate of China's aircraft leasing industry has increased by 20% and decreased by 20%, the simulated values of the lease volume and market share of China's aircraft leasing companies are generated through VENSIM, with the output shown in Figure 10. The simulation shows that if the average financing cost rate drops by 20%, market share will rise from 69.65% to 71.60% in 2025. Reducing the financing cost of the aircraft leasing industry will significantly enhance its market competitiveness and expand its market share.

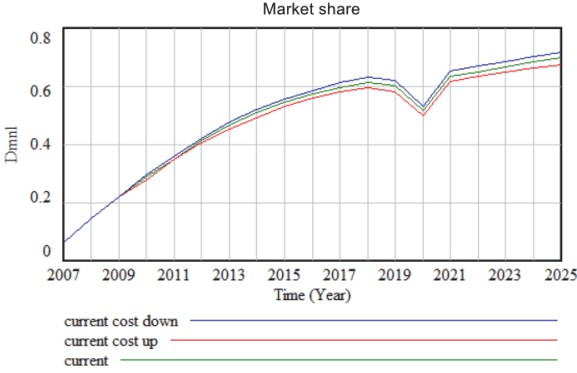 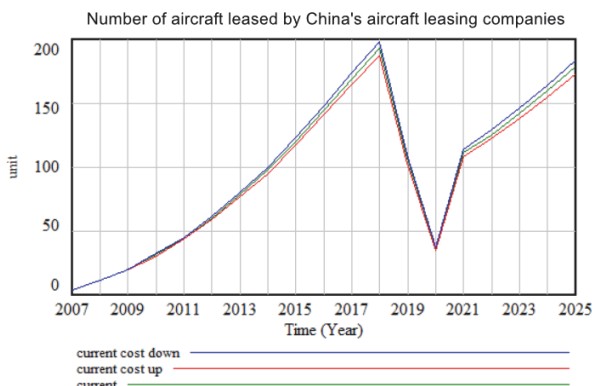

**Figure 10.** The influence of aircraft leasing companies' average financing cost rate on market share and lease volume.

## 6. Discussion

In the present study, we used system dynamics to study the sustainable development of the aircraft leasing industry. We simulated the impact of key variables from the economic, market, enterprise, and environmental subsystems, then established a ranking of the key variables that promote the sustainable development of the aircraft leasing industry to provide effective support for policy-making. Forecasting the growth of the aircraft leasing industry has become difficult amid the ongoing impact of COVID-19. Through simulation, we predicted the development of China's aircraft leasing industry from 2021 to 2025 and solved this problem.

We obtained the quantitative effect of the influence of factors such as GDP growth rate, income tax rate, etc., solving the problem given the current research is mainly qualitative with insufficient quantitative analysis, and improving the pertinence and effectiveness of policy recommendations. We identified that changes in GDP growth have a significant impact on the sustainable development of China's aircraft leasing industry. As the supply capacity of domestic aircraft leasing companies cannot meet market needs, with a sharp increase in GDP growth rate, the market share of Chinese aircraft leasing companies will decline. In this context, foreign aircraft leasing companies will fill the market demand. We found that reducing financing costs and tax rates will help Chinese aircraft leasing companies promote their market share. This is consistent with the conclusions of previous studies. Quantitative research on talent investment and risk control was previously relatively insufficient. Through simulation, we also found that increasing investment in talent and controlling risk better promotes the development of the industry. On this basis, we found that of the positive factors affecting the sustainable development of the aircraft leasing industry, the order of importance from high to low is tax policy, talent investment, financing costs, innovation, and the number of aircraft leasing companies.

As we know, the development of the aircraft leasing industry is facing more and more challenges due to COVID-19. Promoting this development is a practical and urgent problem. We applied the method of system dynamics to study this complex problem and established a model to examine the key measures to promote sustainable development of the aircraft leasing industry. However, there are some limitations in this study. First, we proposed the main factors from four sub-systems, but these factors may not be comprehensive enough. For example, the legal, regulatory, accounting, aircraft supply capacity, and other factors involved in the sustainable development of the aircraft leasing industry are difficult to quantify, and these factors have been simplified during modeling. Second, the parameters in the model were obtained by simple and multiple regression. Third, we designed the supply of funds based on money stock M2, which cannot fully reflect the supply of funds in the financial market. In future research, we will explore better methods for parameter determination that can further optimize the system model. As COVID-19 also influences the aircraft production supply chain, the impact of aircraft supply capacity on the sustainable

development of the aircraft leasing industry needs further research. In addition, with the recognition of financial leasing by airlines, the lease penetration will change. It will affect the sustainable development of the industry. Its impact also needs further research. During the processing of model variables, we also discovered some new problems. For example, it is hard to know how long the coronavirus pandemic will last. Will Airbus and Boeing still be the only two aircraft manufacturers in the next five years? Additionally, in terms of the applicability of the results, and given the varying development levels of each country, further research is needed to determine whether our conclusions can be applied to other developing countries.

From the simulation results, we showed that although the Boeing 737 MAX crisis and COVID-19 have seriously influenced the development of China's aircraft leasing industry in the past two years, the aircraft leasing industry will resume its growth with the gradual recovery of China's economy. In 2021, the market share of Chinese aircraft leasing companies was predicted to return to the value of 63.43% and will gradually rise to 69.6% in 2025. At that time, the number of new aircraft leases in China will return to the 2018 level. This means that the impact of COVID-19 on the aircraft leasing industry has been well controlled.

## 7. Conclusions and Policy Recommendations

This research aimed to identify effective strategies for promoting the sustainable development of China's aircraft leasing industry. Based on a quantitative and qualitative analysis of the SD model, it can be concluded that reducing the average financing cost, reducing the income tax rate, increasing the investment in talent, and controlling risk are important factors to consider when developing effective policies. The results indicate that among the positive factors, the order of importance from high to low is tax policy, talent investment, financing costs, innovation, and the number of aircraft leasing companies. Based on the above conclusions, to promote the sustainable development of China's aircraft leasing industry, the following recommendations are put forward:

For Chinese aircraft leasing companies, first, in terms of business development, it is recommended to focus more on airlines with higher credit ratings and distribute their aircraft leasing business to countries less affected by the COVID-19 pandemic, as the economies of these countries recover faster. Second, in terms of aircraft fleet structure, increase the proportion of mainstream leading aircraft and young aircraft assets, and also the narrow-body aircraft with a higher utilization rate, to comply with the requirements of global green development. New technology aircraft can help airlines save 20–30% of fuel consumption. In the future, the demand for technical iterations of airline stock aircraft will significantly increase and some old technology models will accelerate their exit from the market. Third, increase investment in innovation and talent. The theory of industrial growth has shown that innovation is the key to the transformation of industrial growth from quantitative development to qualitative development. In the sustainable development of China's aircraft leasing industry, innovation-driven is the only way to achieve high-quality development. Since employees in the aircraft leasing industry are usually required to be familiar with finance, aviation, law, taxation, risk control, etc., they are compound talents, which are currently insufficient. The government, universities, and enterprises should strengthen cooperation and jointly cultivate more professionals.

For the government, first, expand the financing channels of aircraft leasing companies, encourage qualified aircraft leasing companies to raise funds through the capital market, and encourage them to issue USD bonds in the global market. Promote aircraft leasing companies to carry out pilot projects of cross-border capital pools in RMB and USD, simplify the management of foreign exchange settlement and payment, and facilitate the adjustment and collection of domestic and foreign currency funds. Encourage aircraft leasing companies to carry out lease asset securitization, encourage more institutions such as trust companies and asset management companies to participate in asset transactions, and expand the scale of aircraft lease assets transactions. Second, give further tax incentives

to the aircraft leasing industry. The change in tax rate has an obvious impact on the income of aircraft leasing projects. It is the main reason that Chinese aircraft leasing companies set up their SPV companies in countries or regions with lower tax rates to carry out aircraft leasing business. Consider Ireland as an example. Its income tax rate is only 12.5%, which is significantly lower than the average level of 25% in other countries. Therefore, Ireland has attracted many aircraft leasing companies to invest there, giving Irish aircraft leasing companies a market share of more than 50% in the global aircraft leasing market. Third, regulate and guide the development of the aircraft leasing industry, avoid vicious competition, and control market risk in the industry. Further, improve the relevant laws and regulations of the aircraft leasing industry, and encourage the aircraft leasing companies to continuously strengthen their risk control ability, operate steadily, and achieve sustainable development.

**Author Contributions:** W.L. and J.L. conceived and designed the experiments; W.L. and L.X. performed the experiments and analyzed the data; J.Z. and L.X. contributed analysis tools; and W.L. and J.L. wrote the paper. All authors have read and agreed to the published version of the manuscript.

**Funding:** This research received no external funding.

**Institutional Review Board Statement:** Not applicable.

**Informed Consent Statement:** Not applicable.

**Data Availability Statement:** Publicly available datasets were analyzed in this study. This data can be found here: http://www.caac.gov.cn/XXGK/XXGK/TJSJ/index_1214.html (accessed on 4 December 2021). http://www.pbc.gov.cn/diaochatongjisi/116219/index.html (accessed on 4 December 2021). https://data.stats.gov.cn/ (accessed on 4 December 2021).

**Conflicts of Interest:** The authors declare no conflict of interest.

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
