# Peer review of "Research on the Sustainable Development and Dynamic Capabilities of China’s Aircraft Leasing Industry Based on System Dynamics Theory"

_sustainability, doi:10.3390/su14031806_

Round 1
Reviewer 1 Report
- Interesting topic, thank you
- Please add more references - some of the statements/facts are not covered with citations (f.ex. introduction) - this will improve the scientific soundness of your work
- Would it be interesting to add the concept of "dynamic capabilities" into the discussion of the topic?
- It is recommended to rework the introduction according to the guidleines of the journal (also including the research question [next to research gap] as well as the structure of the paper)
- Proofreading of the whole article is suggested (see f.ex. line 92)
- Please add sources for the development of figure 1 (as well as figure 2)
- Is this a c.p. diagram? If so, please state that.
- How many years into the future a system dynamic approach is really applicable? Please elaborate
- Very good practical implications
- Please improve the formatting (for the bibliography for example)
Author Response
Dear editor,
We would like to express our sincere thanks to you for the further comments and suggestions on our paper. We have revised our paper based on these comments and suggestions. Our treatment is discussed in the appendix (point-to-point and response). We have also highlighted the changes in the manuscript.
Based on your comments, we have provided the source files of the figures in the article, and the link addresses are as follows:
https://wx.mail.qq.com/ftn/download?func=3&key=c9cd5b63f03cb365fbbe1b63333064668b3a236331306466131d5052530105560f5418070903074b020a05501c085c07051f56500904015304015657065179664547461750590a07545b590a4549495703030d5006004934071c4702432432a007d2a8bd2c182e757079d6344b6ca8dc8166&code=625c10df&k=c9cd5b63f03cb365fbbe1b63333064668b3a236331306466131d5052530105560f5418070903074b020a05501c085c07051f56500904015304015657065179664547461750590a07545b590a4549495703030d5006004934071c4702432432a007d2a8bd2c182e757079d6344b6ca8dc8166&fweb=1&cl=1 If you have any questions, please feel free to contact me. Kind regards. Lin Weiwei

Reviewer 2 Report
When reviewing scientific papers for publication, I usually start with a general overview in terms of a structure, abstract, literature review, methodology, findings of the research, discussion, conclusions, as well as limitations of the study. I also pay attention to a language level, especially if the paper is written in English, and English is not the native language.
The reviewed paper entitled “Research on the sustainable development of China's aircraft leasing industry based on system dynamics theory” is generally structured in a proper way. There are, however no sections ‘Discussion’, ‘limitations of the study’, and ’future directions of the research”. These sections should be added too, given this is a research paper.
In general:
Discussion is an interpretation of the results – implications, significance of results.
- Provide the response to the research question(s)
- Interpret results taking into account alternative explanations - where applicable
- What are the practical implications (and theoretical –where applicable) suggested by the results of your research.
- Include all limitations: this does not weaken your study, but adds to your credibility
- Future directions for research (incompletely answered questions) often derived from limitations.
- New questions which emerge from your research
- Be careful not to “go beyond” your data and results, in particular if the focus of your study is narrow
- You can “suggest”, or even “speculate” in the discussion, but it must be clearly evident what is derived from a result and what is your suggestion, comment or speculation, ...
- You may include a comparison with results of other similar/ compatible studies – if applicable.
Conclusion is the last part of the discussion or a separate chapter:
You may briefly summarize main results (if you haven‘t done this in the Discussion)
Bring the reader back to the research question – concluding with a larger and richer view of the problem/ question under investigation
Authors may add their own opinion and a broader comment of the results, their suggestions, recommendations, ...
Authors may add their proposals, suggestions, recommendations, evaluations, based on the results of the study - if appropriate as a separate chapter or subchapter.
The literature review is average but is strongly founded in the existing literature of the topic. Generally I claim that Author (s) provide solid theoretical foundations for the analysis using appropriate references. I would, however, recommend to add some references devoted to the latest literature associated with the topic in question (including Scopus and Web of Science papers).
Author Response
Dear editor,
We would like to express our sincere thanks to you for the further comments and suggestions on our paper. We have revised our paper based on these comments and suggestions. Our treatment is discussed in the appendix (point-to-point and response). We have also highlighted the changes in the manuscript.
Based on your comments, we have provided the revised paper with the following link:
https://wx.mail.qq.com/ftn/download?func=3&key=9c9d5b39a03ab134aeee1b396336663788b6203961366637464d5108030e50515306180b5600031a5706070b4c0f5601504f540d075556035457055a550e7b371017464d005f0856010b5950154f4b0656530d0a56064b65514c47581322f3a0d9421d0ed9e87fc1d3a65a3675fb2ac62d93&code=cb59a6f7&k=9c9d5b39a03ab134aeee1b396336663788b6203961366637464d5108030e50515306180b5600031a5706070b4c0f5601504f540d075556035457055a550e7b371017464d005f0856010b5950154f4b0656530d0a56064b65514c47581322f3a0d9421d0ed9e87fc1d3a65a3675fb2ac62d93&fweb=1&cl=1
If you have any questions, please feel free to contact me. Kind regards. Lin Weiwei

Round 2
Reviewer 1 Report
Dear authors,
thank you for your revision.
c.p. stands for ceteris paribus and describes the state if the model at hand is not taking outside parameters and variables (which are changing and flexible) into consideration.
All the best for your future research!
Author Response
Dear editor,
Thank you for your patience and explanation for “ceteris paribus”. We have revised our paper according to your comments and suggestions.
Point 1: Is this a c.p. diagram? If so, please state that.
c.p. stands for ceteris paribus and describes the state if the model at hand is not taking outside parameters and variables (which are changing and flexible) into consideration.
Response 1: Your comments are full of preciseness. About the figure 1, yes, it is ceteris paribus diagram. We have stated it in the paper according to your further comments. We have also highlighted the changes in the manuscript.
Reviewer 2 Report
Dear author, Thank you very much for the corrections and additions, which I appreciate. Unfortunately, the "Discussion" section is still a weak point in your article.
Discussion is an interpretation of the results - implications, significance of results.
- Provide the response to the research question (s)
- Interpret results taking into account alternative explanations - where applicable
- What are the practical implications (and theoretical –where applicable) suggested by the results of your research.
- Include all limitations: this does not weaken your study, but adds to your credibility
- Future directions for research (incompletely answered questions) often derived from limitations.
- New questions which emerge from your research
- Be careful not to “go beyond” your data and results, in particular if the focus of your study is narrow
- You can “suggest”, or even “speculate” in the discussion, but it must be clearly evident what is derived from a result and what is your suggestion, comment or speculation, ...
- You may include a comparison with results of other similar / compatible studies - if applicable.
Author Response
Dear editor,
Thank you for the further comments and suggestions on our paper. We have revised our paper according to your comments and suggestions. Please check the attachment.
Kind regards. Lin Weiwei
